# Primary Hyperparathyroidism in Multiple Endocrine Neoplasia Type 2A in Denmark 1930–2021: A Nationwide Population-Based Retrospective Study

**DOI:** 10.3390/cancers15072125

**Published:** 2023-04-02

**Authors:** Magnus Holm, Peter Vestergaard, Morten Møller Poulsen, Åse Krogh Rasmussen, Ulla Feldt-Rasmussen, Mette Bay, Lars Rolighed, Stefano Londero, Henrik Baymler Pedersen, Christoffer Holst Hahn, Klara Bay Rask, Heidi Hvid Nielsen, Mette Gaustadnes, Maria Caroline Rossing, Anne Pernille Hermann, Christian Godballe, Jes Sloth Mathiesen

**Affiliations:** 1Department of ORL Head & Neck Surgery and Audiology, Odense University Hospital, 5000 Odense, Denmark; ma.holm@hotmail.com (M.H.); mette.bay@rsyd.dk (M.B.); christian.godballe@rsyd.dk (C.G.); 2Steno Diabetes Center North Denmark, Aalborg University Hospital, 9100 Aalborg, Denmark; p-vest@post4.tele.dk; 3Department of Endocrinology and Internal Medicine, Aarhus University Hospital, 8200 Aarhus, Denmark; morpouls@rm.dk; 4Department of Endocrinology and Metabolism, Copenhagen University Hospital, 2100 Copenhagen, Denmark; aase.krogh.rasmussen@regionh.dk (Å.K.R.); ulla.feldt-rasmussen@regionh.dk (U.F.-R.); caroline.maria.rossing@regionh.dk (M.C.R.); 5Department of Clinical Medicine, Faculty of Health and Medical Sciences, University of Copenhagen, 2200 Copenhagen, Denmark; 6Department of ORL Head & Neck Surgery, Aarhus University Hospital, 8200 Aarhus, Denmark; larsroli@rm.dk (L.R.); stelon@rm.dk (S.L.); 7Department of ORL Head & Neck Surgery, Aalborg University Hospital, 9100 Aalborg, Denmark; hebp@rn.dk; 8Department of ORL Head & Neck Surgery and Audiology, Rigshospitalet, Copenhagen University Hospital, 2100 Copenhagen, Denmark; christoffer.holst.hahn.01@regionh.dk (C.H.H.); klara.bay.rask@regionh.dk (K.B.R.); 9Department of Clinical Biochemistry, Zealand University Hospital, 4000 Roskilde, Denmark; hhne@regionsjaelland.dk; 10Department of Molecular Medicine, Aarhus University Hospital, 8200 Aarhus, Denmark; mette.gaustadnes@clin.au.dk; 11Center for Genomic Medicine, Copenhagen University Hospital, 2100 Copenhagen, Denmark; 12Department of Endocrinology, Odense University Hospital, 5000 Odense, Denmark; pernille.hermann@rsyd.dk; 13Department of Clinical Research, University of Southern Denmark, 5000 Odense, Denmark

**Keywords:** multiple endocrine neoplasia type 2A, primary hyperparathyroidism, hypercalcemia, *RE*arranged during *T*ransfection, national cohort, Denmark

## Abstract

**Simple Summary:**

Multiple endocrine neoplasia 2A (MEN 2A) is a rare hereditary cancer syndrome, in which primary hyperparathyroidism (PHPT) is reported in up to 35% of affected individuals. Recent studies suggest a lower frequency and a milder course of PHPT in MEN 2A, but these studies often lack a strict definition of PHPT and are frequently carried out at smaller research centers. This could result in diverging and incomplete data. Consequently, we aimed to investigate PHPT in a complete nationwide cohort of MEN 2A to suggest a representative frequency and clinical course of PHPT. This may alter the information given to MEN 2A patients by their caretakers on the likelihood of developing PHPT and on the clinical course of PHPT.

**Abstract:**

Studies of primary hyperparathyroidism (PHPT) in multiple endocrine neoplasia type 2A (MEN 2A) shows divergence in frequency, disease definition, reporting of clinical characteristics and traces of selection bias. This is a nationwide population-based retrospective study of PHPT in MEN 2A, suggesting a representative frequency, with complete reporting and a strict PHPT definition. The Danish MEN 2A cohort 1930–2021 was used. Of 204 MEN 2A cases, 16 had PHPT, resulting in a frequency of 8% (CI, 5–12). Age-related penetrance at 50 years was 8% (CI, 4–15). PHPT was seen in the American Thyroid Association moderate (ATA-MOD) and high (ATA-H) risk groups in 62% and 38% of carriers, respectively. Median age at PHPT diagnosis was 45 years (range, 21–79). A total of 75% were asymptomatic and 25% were symptomatic. Thirteen underwent parathyroid surgery, resulting in a cure of 69%, persistence in 8% and recurrence in 23%. In this first study with a clear PHPT definition and no selection bias, we found a lower frequency of PHPT and age-related penetrance, but a higher age at PHPT diagnosis than often cited. This might be affected by the Danish *RET* p.Cys611Tyr founder effect. Our study corroborates that PHPT in MEN 2A is often mild, asymptomatic and is associated with both ATA-MOD and ATA-H variants. Likelihood of cure is high, but recurrence is not infrequent and can occur decades after surgery.

## 1. Introduction

Multiple endocrine neoplasia type 2 (MEN 2) is a rare hereditary cancer syndrome caused by germline variants in the *RE*arranged during *T*ransfection (*RET*) proto-oncogene [1,2,3,4,5] and is subdivided into MEN 2A and MEN 2B, with a point prevalence of 13–24 per million and 0.9–1.65 per million [6,7,8,9], respectively. MEN 2A is associated with medullary thyroid carcinoma (MTC) in almost all cases, pheochromocytoma (PHEO) in up to 42% of cases, primary hyperparathyroidism (PHPT) in up to 35% of cases, and more rarely cutaneous lichen amyloidosis and Hirschsprung’s disease [10].

Recent studies on MEN 2A show a PHPT frequency of 2–18% [11,12,13,14,15,16,17,18,19,20,21], which is relatively lower than the frequency of 19–35% found in larger single- and multicenter studies from the 1990s [22,23,24,25,26,27]. However, most of these recent studies are smaller single-center studies [11,13,14,15,16,17,18,19,21], making them vulnerable to selection bias. To our knowledge, only three nationwide studies have reported the frequency of PHPT in MEN 2A [6,28,29]. However, all of the studies lack a definition of PHPT. One study included only index cases [28], while another study included less than 1/5 of the expected total MEN 2 population [29]; all of which providing uncertainty to their frequency figures. Additionally, age-related penetrance of PHPT in MEN 2A has never been investigated in an unselected national cohort, but instead mostly in specific variants or exons [12,20,22]. It is also noted that both older and more recent studies define PHPT in MEN 2A as either “biochemical or histological” [14,25,27,30,31,32], “biochemical and histological” [11,12,22,23,26], or “biochemical with normalization after surgery” [16,21]. The different methods could affect MEN 2A PHPT data. Furthermore, only two single-center studies have consistently reported all the essential characteristics (frequency of PHPT, sex, age at diagnosis, symptoms, biochemistry, *RET* variants, histopathology, glands affected, the timing of diagnosis, surgical procedures, postoperative outcome, and complications) regarding PHPT in MEN 2A [19,26]. 

Consequently, we aimed to estimate the frequency and age-related penetrance of PHPT in MEN 2A and report all essential characteristics of PHPT in a nationwide population-based MEN 2A cohort with minimum selection bias and a strict PHPT definition.

## 2. Materials and Methods

### 2.1. Study Design and Setting

This investigation is a nationwide population-based retrospective study of all patients born after 1 January 1930 and recognized with MEN 2A in Denmark before 1 April 2021. Special attention is given to the subgroup (n = 16) diagnosed with PHPT before the last serum calcium measurement.

### 2.2. Data Sources

#### 2.2.1. Danish MEN 2 Cohort 1701–2014

Between 1 January 1701, and 31 December 2014, 262 cases (250 MEN 2A, 12 MEN 2B) were registered in the Danish MEN 2 cohort [7,8]. This cohort was created using the Danish *RET* cohort 1994–2014, pedigree linkage (based on the Civil Registration System and the Danish National Archives), the Danish MTC cohort 1960–2014 (based on the Danish Thyroid Cancer Database, the Danish Cancer Registry, and the Danish Pathology Register), a Danish PHEO cohort, a Danish hyperparathyroidism cohort, a Danish Hirschsprung cohort, a Danish MEN 2A cohort (all included from 1968–2014 and based on the Danish Pathology Register and the Danish National Patient Registry), a Danish lichen amyloidosis cohort 1976–2014 (based on the Danish National Patient Registry), a systematic literature search (Cochrane, Embase, PubMed, Scopus and Web of Science) and a nationwide collaboration of all Danish MEN 2 treatment centers. This has been meticulously described elsewhere [7,8,33].

#### 2.2.2. Danish RET Cohort 2015–2021

This cohort consists of all Danish inhabitants who had undergone *RET* testing between 1 January 2015, and 1 April 2021 and is an extension to two previously described Danish *RET* cohorts 1994–2014 and 1994–2017 [33,34]. In the Danish *RET* cohort 2015–2021 57 cases tested positive for a pathogenic *RET* variant [35]. Screening covered the entire country and was conducted by the Center of Genomic Medicine at Copenhagen University Hospital, Department of Molecular Medicine at Aarhus University Hospital, and Department of Clinical Biochemistry at Zealand University Hospital. One Danish inhabitant was screened in the UK but shortly after moved permanently to Denmark and was included. Another patient tested positive for a pathogenic *RET* variant between 2015 and 2021 in Denmark but was already recognized as having MEN 2 in the Danish MEN 2 cohort 1701–2014, and was therefore excluded from the Danish *RET* cohort 2015–2021. Thus, the 2015–2021 cohort includes 57 new MEN 2 cases.

#### 2.2.3. Pedigree Linkage

Through genetic workup, one of the index cases from the Danish *RET* cohort 2015–2021 was linked to an already-known Danish MEN 2 family through two generations. This provided two new MEN 2 cases recognized as obligate carriers.

#### 2.2.4. Medical Records

Where data from the abovementioned sources were insufficient, medical records were used.

### 2.3. Participants

The Danish MEN 2 cohort 1701–2014 included 262 cases. Of these, one had erroneously been reported to carry the p.Cys611Tyr variant, when in reality carried *RET* wildtype. In 21 of the 262 cases, the MEN 2 diagnosis was disregarded as the pedigree linkage was considered too weak (cases where multiple generations separated a likely, but genetically untested MEN 2 patient from a MEN 2 patient that had tested positive for a pathogenic *RET* germline variant). After exclusion of these 22 cases, the cohort comprised 240 cases. From the Danish *RET* cohort 2015–2021 (n = 57) and following pedigree linkage (n = 2), 59 new cases were added. This yielded 299 cases, of which 36 of the patients were born between 1701 and 1900. Thus, the updated Danish MEN 2 cohort 1901–2021 contained 263 cases born and recognized with MEN 2 from 1901–2021. Of these, 34 were born 1901–1929 and excluded. Among the remaining 229 cases, 13 had MEN 2B and were excluded from this study. Consequently, an updated Danish MEN 2A cohort 1930–2021 contained 216 cases. Of these, 11 cases never had their calcium level measured as they declined further MEN 2A workup after positive *RET* testing (n = 3), as their MEN 2A diagnosis was unknown at the last follow-up (death/emigration) (n = 4), or as they were too young for PHPT screening (n = 4). Unfortunately, data were missing for one case. This left us with a cohort of 204 cases (Figure 1).

Inclusion was chosen from 1 January 1930, under the assumption that virtually all patients with MEN 2A in Denmark born after this date have been captured. Since MTC usually is the first manifestation of MEN 2A and node-negative MTC often develops between the third to fifth decade of life, practically all patients with MTC as part of MEN 2A should have been recognized in the Danish MTC cohort 1960–2014 [29,36,37,38]. Additionally, the incidence of MEN 2A in Denmark has been roughly stable since the 1930s, arguing that virtually all MEN 2A patients since then have been captured [7].

### 2.4. Variables

PHPT was defined as hypercalcemia and an elevated or inappropriately normal parathyroid hormone level in accordance with international guidelines [39]. Symptoms included non-traumatic fractures, osteoporosis, nephrolithiasis, nephrocalcinosis, reduced renal function, polyuria, polydipsia, peptic ulcer, constipation, pancreatitis, dyspepsia, nausea, muscle weakness, and muscle atrophy [40,41]. Hypercalcemia was divided into Grade 1–5 in keeping with Common Terminology Criteria for Adverse Events (CTCAE) version 5.0 [42]. 

Cure was defined as the reestablishment of normal calcium homeostasis lasting for a minimum of six months after parathyroidectomy (PTX). Persistence was considered as failure to achieve normocalcemia within six months of PTX, while recurrence was deemed as recurrence of hypercalcemia after a normocalcemic interval at more than six months after PTX [43].

Permanent hypoparathyroidism was defined as the use of active D vitamin (alfacalcidol, ATC A11CC03; calcitriol, ATC A11CC04) for more than one year after PTX [44,45,46]. Permanent recurrent laryngeal nerve palsy was defined as vocal cord palsy determined by laryngoscopy (flexible transnasal fiber-optic laryngoscopy or indirect mirror laryngoscopy) that persisted for more than one year after surgery.

C-cell hyperplasia, MTC, PHEO, cutaneous lichen amyloidosis, and Hirschsprung’s disease were considered when diagnosed histologically. Synchronous occurrence of diseases of MEN 2A was defined as suspicion of two or more MEN 2 manifestations (MTC, PHEO, PHPT, cutaneous lichen amyloidosis, Hirschsprung’s disease) at the same time. In this context, before and after was defined as suspicion of one MEN 2 manifestation before or after suspicion of another MEN 2 manifestation.

*RET* variants were grouped according to the MTC risk classification in accordance with the 2015 American Thyroid Association (ATA) guidelines on MTC [47].

Age at last follow-up was calculated from date of birth until date of last serum calcium measurement.

### 2.5. Statistical Analysis

Continuous data were displayed as median and range. Categorical data were shown with absolute and relative values. The Kaplan–Meier method was used for estimating the age-related penetrance of PHPT. All analyses were made using STATA 17.0 (StataCorp, College Station, TX, USA).

## 3. Results

### 3.1. Overall MEN 2A Cohort

A total of 204 MEN 2A patients were included in this study. The distribution of variants was p.Cys611Tyr (59%), p.Leu790Phe (14%), p.Cys618Tyr (5%), p.Cys620Arg (5%), p.Cys618Phe (4%), p.Cys634Arg (4%), p.Cys611Trp (3%), p.Val804Met (1%), p.Cys634Tyr (1%), p.Val804Leu (1%), p.Asp631Tyr (0.5%) p.Cys634Tyr + p.Leu791Phe (0.5%) and p.Ser891Ala (0.5%).

#### 3.1.1. Frequency

Of the 204 patients, 16 had developed PHPT before the last serum calcium measurement, resulting in a frequency of 8% (CI, 5–12). In the ATA moderate (MOD) and high (H) groups, frequencies were 5% (CI, 3–9) and 50% (CI, 21–79), respectively (*p* < 0.001, Fisher’s exact test).

#### 3.1.2. Age-Related Penetrance

The age-related penetrance of PHPT at 20 years was 0% (CI, 0–0), at 30 years 1% (CI, 0–5), at 40 years 4% (CI, 1–8), at 50 years 8% (CI, 4–15), at 60 years 14% (CI, 8–23), at 70 years 16% (CI 10–26) and at 80 years 25% (CI, 12–50) (Figure 2). There was a significantly higher hazard ratio of 9 (CI, 3–25) in age-related penetrance for the ATA-H group when compared to the ATA-MOD group (*p* < 0.001, Cox regression).

### 3.2. PHPT Cohort

#### 3.2.1. Demographics

Descriptive data of the 16 patients are shown in Table 1. The cohort consisted of 11 men (69%) and five women (31%), with a female-to-male ratio of 0.45. The most frequent variant was p.Cys611Tyr (50%), followed by p.Cys634Arg (38%), p.Cys618Phe (6%), and p.Leu790Phe (6%). Median age at MEN 2A diagnosis was 42 years (range, 4–82), while median age at PHPT diagnosis was 45 years (range, 21–79). Median follow-up after PHPT diagnosis was 10 years (range, 0.5–29).

#### 3.2.2. Clinical Characteristics

In cases with pertinent data (n = 15), the median ionized calcium level was 1.38 mmol/L (range, 1.33–1.79). In the last patient, PHPT diagnosis was based on an elevated total calcium level. Median PTH (n = 16) was 9.3 pmol/L (range, 3.1–18.8). No patients had hypercalcemic crises. 

A total of 75% were asymptomatic and 25% were symptomatic at PHPT diagnosis; The most frequent symptoms were osteoporosis (n = 2) and polydipsia (n = 2). Other symptoms included nephrolithiasis, polyuria and nausea. Median age at PHPT diagnosis in the asymptomatic group was 42 years (range, 21–63) and 55 years (range, 39–79) in the symptomatic group (*p* = 0.23, Mann–Whitney–Wilcoxon-rank test).

#### 3.2.3. Surgical Characteristics

Thirteen (81%) of 16 patients underwent parathyroid surgery. Of these, nine (69%) had subtotal (sub.) PTX (three of which also had autotransplantation), three (23%) had selective (sel.) PTX, and one (8%) had a neck exploration without PTX. Nine patients (69%) had simultaneous total thyroidectomy and parathyroid surgery, while four (31%) had parathyroid surgery after thyroidectomy. Three patients had no parathyroid surgery before the last follow-up. In these cases, the patients had developed PHPT several years after total thyroidectomy, and a multidisciplinary team had assessed that surgical treatment and risk of complications would not outweigh potential benefits. Thus, all three patients are treated conservatively.

Histopathology showed adenoma in six patients (50%), hyperplasia in five (42%), and the occurrence of both in one patient (8%). No patients had parathyroid carcinoma.

Among those operated, nine (69%) were cured after a median follow-up of 11 years (range, 0.3–28), one (8%) had persistence, and three (23%) had recurrence 3, 13 and 20 years after first PTX (Figure 3).

Six (46%) had permanent hypoparathyroidism after the first parathyroid surgery. All six had simultaneously undergone total thyroidectomy and sub. PTX. No patients had postoperative recurrent laryngeal nerve palsy.

#### 3.2.4. Thyroid

All 16 patients had undergone thyroidectomy. Histopathology showed MTC in 69% (11/16) and C-cell hyperplasia in 31% (5/16). MTC was diagnosed synchronously with PHPT in 55% (6/11) and before PHPT in 45% (5/11) (range, 17–39 years). Similarly, C-cell hyperplasia was diagnosed synchronous with PHPT in 60% (3/5) and before PHPT in 40% (2/5) (range, 16–17 years). All patients with C-cell hyperplasia were asymptomatic of PHPT at PHPT diagnosis.

#### 3.2.5. PHEO

PHEO was diagnosed in 56% (9/16). PHEO was synchronous with PHPT in 56% (5/9) and metachronous in 44% (4/9) patients: PHEO preceded PHPT in 22% (2/9) and followed PHPT in 22% (2/9). No patients had cutaneous lichen amyloidosis or Hirschsprung’s disease.

## 4. Discussion

In this national retrospective cohort study of 204 patients with MEN 2A from 1930 to 2021 in Denmark, we characterized the 16 patients who developed PHPT in terms of demographical, clinical, and surgical characteristics. We had a PHPT frequency of 8% and a median follow-up after PHPT diagnosis of 10 years. 

### 4.1. Limitations

We did not have data on the calcium level in 12 cases (6%) from the overall cohort. Four carried the *RET* p.Cys611Tyr variant and were 2–3 years old at the last follow-up and therefore below the recommended age for PHPT screening in the ATA-MOD group, starting at age 16 years [47]. Combined with the fact that the hitherto youngest p.Cys611Tyr carrier reported with PHPT was 40 years at diagnosis, one might methodologically justify including the four cases in our overall cohort as not having PHPT [48]. This would not change the PHPT frequency but reduce the number of missings to 4%. Three declined further MEN 2A workup after testing positive for the *RET* p.Leu790Phe variant at ages 5, 10, and 14 years. To our knowledge, no carrier of this variant has previously been described with PHPT [35]. Consequently, excluding the three p.Leu790Phe carriers, our PHPT frequency has likely not decreased. Among the remaining five cases, one p.Cys611Trp carrier emigrated at 19 years, one p.Cys618Tyr carrier emigrated at 31 years, one p.Cys611Tyr carrier emigrated at 35 years and two other p.Cys611Tyr carriers died at 45 years and 52 years. In four of these five cases, MEN 2A was undiagnosed at the last follow-up. Due to the relatively young age at the last follow-up and the fact that the p.Cys611Trp variant has not previously been associated with PHPT [35], these carriers were unlikely to have markedly increased the PHPT frequency had they been included. Despite the very small risk that we might have missed a PHPT diagnosis in the 12 cases, we excluded them, for strict methodological reasons, to suggest a representative picture of PHPT in MEN 2A in a nationwide setting.

### 4.2. Overall MEN 2A Cohort

#### 4.2.1. Frequency

Our PHPT frequency (8%) is lower than that reported in several older studies (19–35%) [22,23,24,25,26,27]. The difference might be explained by an excess of the more severe ATA-H phenotype in the older studies. 

Only three other nationwide studies have reported the frequency of PHPT in MEN 2A. One study determines the frequency in a complete national cohort but lacks a PHPT definition and is not a dedicated MEN 2A PHPT study [6]. The same applies in the two other national studies [28,29]. One of these studies was based on voluntary physician questionaries in Japan and included 230 MEN 2 cases. However, considering the number of inhabitants in Japan at the year of study completion (125 million in 1995) [49], and the minimum prevalence of MEN 2 (14 per million) [10], the study needed inclusion of >1500 MEN 2 cases for complete coverage. Therefore, the study carries an inevitable risk of selection bias and the reported PHPT frequency of 12% (25/213) should be interpreted with caution [29]. Caution must also be taken in the other nationwide study reporting a PHPT frequency of 5% (19/366) in France. This frequency was calculated solely from MEN 2A index cases, and full data were only available in 82% (366/448) of cases.

Our frequency of 8% does not vary substantially from the frequency in recent single- and multicenter studies of PHPT in patients with MEN 2A [12,14,15,16,18,19,21] where the median frequency of PHPT is 9.3% (4.2–17.7%). Four of these studies only examine codon 634 carriers and report a PHPT frequency between 5–11% [12,16,18,21]. A collaborative multicenter study by the International *RET* exon 10 consortium found a PHPT frequency of 3% [20], but our cohort also contains ATA-H variants, which may explain a higher PHPT frequency. When including only ATA-MOD carriers, our frequency was 5%, while the frequency in ATA-H carriers was 50%. Thus, there seems to be a genuine difference in frequency of PHPT between the two groups. A recent study reports a frequency of 18% [14] but defines PHPT both “biochemically or histologically”, resulting in a broader definition of PHPT. Hence, this result should be interpreted cautiously since this definition could overestimate the frequency compared to our strictly biochemical PHPT definition. Based on the abovementioned difference among the ATA-MOD and ATA-H groups, we cannot rule out that the p.Cys611Tyr (ATA-MOD) founder effect may lower our PHPT frequency. However, our results regarding PHPT frequency resemble those from recent studies, including a high proportion of codon 634 carriers [14,15,19] or compromising codon 634 carriers only [12,16,18,21].

#### 4.2.2. Age-Related Penetrance

The age-related penetrance of PHPT in this study is generally lower than that reported in most other studies. This is likely explained by a difference in study cohorts, where our overall cohort consisted of 94% ATA-MOD carriers and 6% ATA-H carriers, while the other multicenter studies included solely ATA-H carriers [12,18,22]. Although not quite comparable, our results mostly resemble those (age-related penetrance of 50% by age 82 years) reported by the International *RET* exon 10 consortium [20].

### 4.3. PHPT Cohort

#### 4.3.1. Demographics

Our female-to-male ratio (0.45) is markedly lower than that (1.2–2.6) reported in other recent [14,32] and older [25,26,27,30,50] large MEN 2A PHPT studies. An explanation might be that the other studies are single- or multicentered, compared to this population-based study that spans nationwide. Another difference between the largest studies (n = 35–67) and the present study is the sample size of MEN 2A patients with PHPT (n = 16) [25,26,27,30]. To our knowledge, no evidence points toward a selection of sex in this autosomal hereditary disease.

We have a relatively high proportion of p.Cys611Tyr carriers (50%) in our PHPT cohort. This corresponds with the proportion of p.Cys611Tyr carriers (59%) in our overall cohort (n = 204), but deviates from other MEN 2A PHPT cohorts as they often include only carriers of codon 634 variants [12,13,16,18,21,22]. Some studies, all single-centered, have an overrepresentation of carriers with codon 634 variants (73–93%) [14,15,19,26]. This deviates from the proportion of codon 634 carriers (38%) in our PHPT cohort. A higher proportion of codon 634 carriers is expected since it is the most commonly affected in MEN 2A [10] and PHPT in MEN 2A [47]. The difference in genotype composition is likely caused by the Danish *RET* p.Cys611Tyr founder effect [51].

Median age at PHPT diagnosis (45 years) is notably higher than almost all median ages (range, 28–41 years) in both older single- and multicenter [25,27,50] and recent single-center studies [14,19,21], but the median age in our study resembles that (46 years) reported in a larger multicenter study [20]. The latter may be expected as the study only included carriers of less aggressive variants (ATA-MOD) located in exon 10. Even though the mean age at PHPT diagnosis (range, 30–42 years) in older [11,22,23,26,30] and more recent [12,16,18] single- and multicenter studies cannot be compared directly to the median age at PHPT diagnosis in this study, almost all seem strikingly lower. Our heterogeneous cohort might explain this contrast in age with an overweight of ATA-MOD variants (62%) compared to the overrepresentation [14,19,26] and reporting only [12,13,16,18,21,22] of ATA-H codon 634 variants. Another reason for this age difference could be that eight of our sixteen patients were 39 years or older when diagnosed with MEN 2A. They were diagnosed with MEN 2A and PHPT concomitantly, and as they had never been screened for PHPT before, it is unknown if PHPT could have been diagnosed earlier. In other similar studies, sparse information on age at MEN 2A diagnosis hinders comparison. Another reason for this difference could be that many studies use a PHPT definition that can be both “biochemical or histological” and many MEN 2A patients are prophylactically thyroidectomized at a young age, which implies that a noticeable number also may have intentional or unintentional PTX. Therefore, they might be histologically diagnosed with hyperparathyroidism without biochemical PHPT, and thus lower the age at PHPT diagnosis [14,25,27,30,32].

The median follow-up time after PHPT diagnosis of 10 years is longer than the median follow-up reported in older studies of 6 [50] and 8 years [25]. Since the last of the two studies defines PHPT both biochemically or histologically, it can cause a longer follow-up time as the same circumstance as the one described above can play out. Most studies report a mean follow-up time and cannot be directly compared to our results [14].

#### 4.3.2. Clinical Characteristics

The mildly elevated ionized calcium level (median 1.38 mmol/L) in this study resembles the results from earlier studies that have complete reports of calcium biochemistry [19,22,25,27,30,32,50]. Our PTH levels also correspond nicely with the few other studies that report PTH levels [24,32]. No patients had hypercalcemic crisis. This is in accordance with most other studies [19,25,30,50]. In studies reporting hypercalcemic crisis, this is a relatively infrequent event among the symptomatic patients (8–13%) [14,22,26,32].

In our cohort, 75% were asymptomatic of PHPT and 25% were symptomatic. This corresponds well with the literature reporting 60–94% being asymptomatic and 6–40% symptomatic [14,19,20,22,24,25,27,30,50]. We found no significant difference in age at PHPT diagnosis between the asymptomatic and symptomatic patients, but a type 2 error cannot be excluded. 

#### 4.3.3. Surgical Characteristics

Studies from the 1990s suggested an operative strategy of total [26] or sub. PTX [23,24] regarding PHPT in MEN 2A. Simultaneously, a considerable number of studies recommended sel. PTX [25,27,50] as they found that recurrence of disease was relatively rare [50] and not related to the extent of resection [25]. The success rate is to be weighed against the risk of permanent hypoparathyroidism, which was associated with the aggressiveness of the operation [27]. Subsequently, more and more studies concluded that sel. PTX of enlarged glands was the appropriate surgical method considering persistence, recurrence, and permanent hypoparathyroidism [11,14,15,21,32]. Only one Japanese study recommended total PTX with autotransplantation [13]. 

A total of 69% of our patients had sub. PTX (three with autotransplantation), 23% had sel. PTX, 0% had total PTX, and 8% had a neck exploration without finding of a parathyroid gland. A similar occurrence of sub. PTX (24–54%) is seen in some early studies [19,25,26,50], but not all (50–87%) [24,26,27,30,50]. In recent studies, sel. PTX is the preferred surgical procedure [14,21,32]. PTX with autotransplantation (7–38%) has been reported in most studies [11,19,24,25,26,27,30]. In our study, three patients did not undergo parathyroid surgery after careful multidisciplinary decision-making. Only few studies report similar cases [19,27]. Thus, further research is needed to evaluate this strategy in MEN 2A.

The results of 42% hyperplasia and 50% adenoma of the parathyroid glands, with one patient exhibiting both, seem to be in accordance with both older single- and multicenter studies [22,25,26,27]. Additionally, recent single-center studies [14,19] report a 50/50 distribution. No patient had parathyroid carcinoma, which is also exceptionally rare in MEN 2A [52].

Our data on persistence (8%) and recurrence (23%) of PHPT resembles large single- and multicenter studies reporting persistence (5–11%) and recurrence (9–14%) [25,26,27].

Almost half of our patients who underwent PTX had permanent hypoparathyroidism. The rate is relatively high compared to the recent literature (13–31%) [14,19,32]. This likely reflects our relatively high rate of patients undergoing sub. PTX (62%). We therefore advocate for sel. PTX as the first surgical approach as also recommended in recent guidelines [47]. None of our patients developed postoperative laryngeal nerve palsy, as is consistent with one study [13], but not all studies (6–14%) [11,32]. These figures point to a relatively low risk of this complication.

#### 4.3.4. Thyroid

All patients had either MTC or C-cell hyperplasia. PHPT did not precede MTC or C-cell hyperplasia in any of our patients. Both disease entities were diagnosed synchronously with PHPT in roughly 60% of cases, and before PHPT in roughly 40% of cases. A similar trend is also found in older single- and multicenter studies where PHPT was diagnosed before MTC in 0–14%, diagnosed synchronously with MTC in 52–88%, and diagnosed after MTC in 0–37% of patients [11,22,23,24,25,26,27,30]. Only two recent single-center studies report on PHPT diagnosis before (0–7%), synchronously (38–60%) or after (33–67%) MTC diagnosis [19,32]. Their results correspond well with our nationwide data.

#### 4.3.5. PHEO

Among the 56% of our patients diagnosed with PHEO, 56% were diagnosed synchronously with PHPT, while 22% were diagnosed before PHPT and 22% after PHPT. There are limited reports of PHEO compared to PHPT in the literature [22,27]. Two studies reported that PHPT preceded PHEO in 3% of cases [22,25]. The diseases in MEN 2A are often diagnosed in a certain order, mostly starting with MTC, followed by PHEO and lastly PHPT [10]. The abovementioned studies and our data support the fact that MEN 2A not always presents in this order, even though this is most common.

## 5. Conclusions

In this first nationwide study with a clear definition of PHPT and no selection bias, we found a lower frequency of PHPT than often cited in the literature. Additionally, we found a lower age-related penetrance and a higher age at PHPT diagnosis. All of which may be affected by the Danish *RET* p.Cys611Tyr founder effect. Finally, our study corroborates that PHPT in MEN 2A often is a mild and asymptomatic disease that is associated with both ATA-MOD and ATA-H variants. Likelihood of cure is high, but recurrence is not infrequent and can occur decades after the first PTX. 

## Figures and Tables

**Figure 1 cancers-15-02125-f001:**
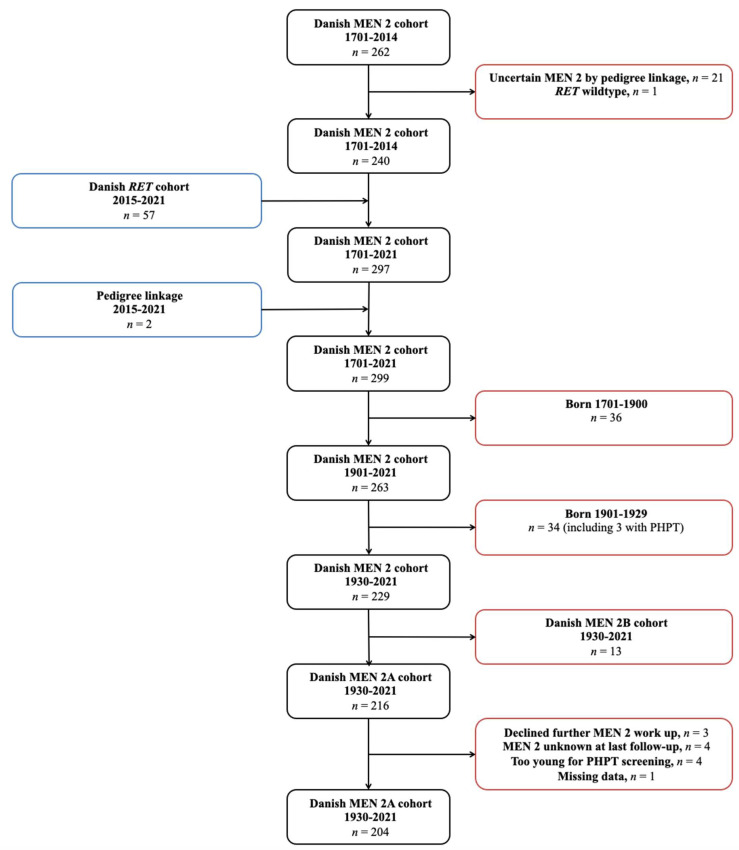
Flowchart showing identification of the Danish MEN 2A cohort 1930–2021. Abbreviations: MEN 2, multiple endocrine neoplasia type 2; *RET*, *RE*arranged during *T*ransfection; PHPT, primary hyperparathyroidism.

**Figure 2 cancers-15-02125-f002:**
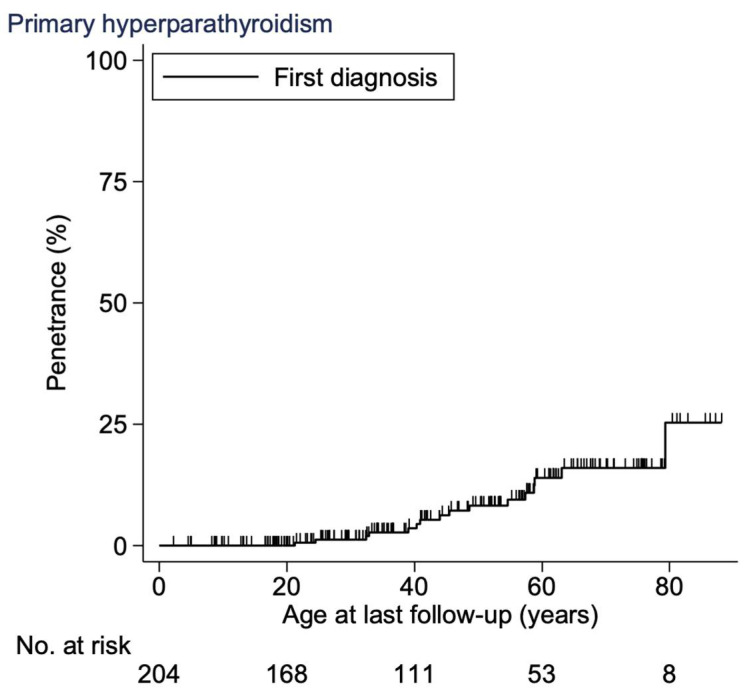
Age-related penetrance of PHPT in the Danish MEN 2A cohort 1930–2021. Abbreviations: PHPT, primary hyperparathyroidism; MEN 2A, multiple endocrine neoplasia type 2A.

**Figure 3 cancers-15-02125-f003:**
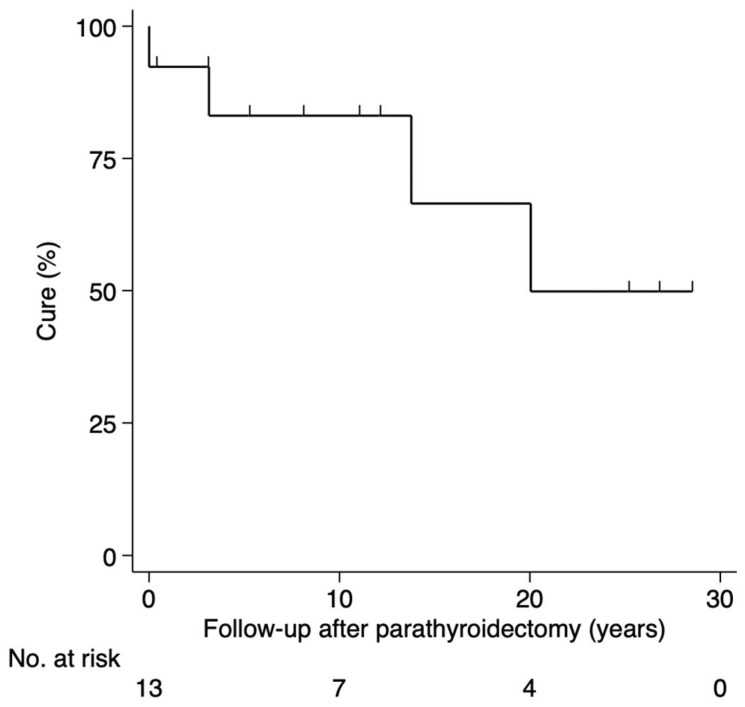
Status after surgery for PHPT in the Danish MEN 2A cohort 1930–2021. Abbreviations: PHPT, primary hyperparathyroidism; MEN 2A, multiple endocrine neoplasia type 2A.

**Table 1 cancers-15-02125-t001:** Characteristics of patients with PHPT from the Danish MEN 2A cohort 1930–2021.

MEN 2A Diagnosis					Thyroid			PHEO, First			PHPT								
Pt No.	Sex	Age, Years	*RET* Variant	Age, Years		Surgery	Pathology	Age, Years	Side	Age, Years		Calcium Severity Grade	Symptoms	Surgery	Pathology	No. Glands Affected	Follow-Up		
																	Age, Years	Status ^c^	Permanent Hypo ^f^
1	F	82	p.Cys634Arg	20		ST	MTC			63		1	N	Sel. PTX	Hyperplasia	2	89	Cured	N
2	M	5	p.Cys611Tyr	6		TTX + SND	CCH			24		1	N				30	Not OP	
3	M	42	p.Cys611Tyr	42		TTX	MTC			40		1	N	Sub. PTX + A	Adenoma	1	69	Cured	N
4	F	33	p.Cys611Tyr	33		TTX + SND	CCH			32		1	N	Sub. PTX + A	Adenoma	1	53/61	Rec/Cured ^d^	Y
5	M	19	p.Cys634Arg	19		TTX	MTC	19	Unilateral	58		1	N				59	Not OP	
6	F	4	p.Cys634Arg	4		TTX + SND	CCH	28	Unilateral	21		1	N	Sel. PTX	Adenoma	1	27/30	Rec/Per ^e^	N
7	M	16	p.Cys634Arg	16		TTX	MTC	21	Unilateral	32		1	N	Sub. PTX	Hyperplasia	2	56	Rec	N
8	M	43	p.Cys634Arg	32		TTX + SND	MTC	43	Bilateral	43		3	N	Exploration		1 ^b^	49	Per	N
9	M	40	p.Cys611Tyr	40		TTX	MTC	56	Unilateral	40		1 ^a^	N	Sel. PTX	Hyperplasia	1	69	Cured	N
10	M	40	p.Cys611Tyr	40		TTX	MTC			57		1	Y				68	Not OP	
11	F	45	p.Cys618Phe	45		TTX + CND	MTC			45		1	N	Sub. PTX + A	Adenoma	1	55	Cured	Y
12	F	39	p.Cys611Tyr	39		TTX + CND	MTC	39	Unilateral	39		2	Y	Sub. PTX	Adenoma	1	50	Cured	Y
13	M	54	p.Cys634Arg	54		TTX + CND	MTC	54	Bilateral	54		2	Y	Sub. PTX	Adenoma/Hyperplasia	3	61	Cured	Y
14	M	58	p.Leu790Phe	58		TTX + CND	CCH			58		1	N	Sub. PTX	Hyperplasia	2	63	Cured	N
15	M	79	p.Cys611Tyr	79		TTX + CND	MTC	79	Unilateral	79		1	Y	Sub. PTX	Hyperplasia	3	81	Cured	Y
16	M	48	p.Cys611Tyr	48		TTX + CND	CCH	48	Unilateral	48		3	N	Sub. PTX	Adenoma	1	49	Cured	Y

**Abbreviations**: MEN 2A, multiple endocrine neoplasia type 2A; ST, subtotal thyroidectomy; TTX, total thyroidectomy; SND, selective neck dissection; CND, central neck dissection; CCH, C-cell hyperplasia; MTC, medullary thyroid carcinoma; PHEO, pheochromocytoma; PHPT, primary hyperparathyroidism; Pt, patient; no, number; *RET*, *RE*arranged during *T*ransfection; M, male; F, female; N, no; Y, yes; Sel. PTX, selective parathyroidectomy; Sub. PTX, subtotal parathyroidectomy; A, autotransplantation; OP, operated; Rec, recurrence; Per, persistence; Hypo, hypoparathyroidism. ^a^ Total albumin corrected s-calcium. ^b^ A pathological parathyroid gland was detected on scintigraphy and at surgery but was not removed due to great risk of complications. ^c^ Cured was defined as the reestablishment of normal calcium homeostasis lasting for a minimum of six months after parathyroidectomy, persistence as failure to achieve normocalcemia within six months of parathyroidectomy, recurrence as recurrence of hypercalcemia after a normocalcemic interval at more than six months after parathyroidectomy [43]. ^d^ Sel. PTX of previously autotransplanted parathyroid at age 59 years. Subsequently cured. ^e^ Sub. PTX + A + thymectomy at age 30 yrs. Subsequently persistence. ^f^ Permanent hypoparathyroidism was defined as the use of active D vitamin (calcitriol, ATC A11CC04, alfacalcidol, ATC A11CC03) for more than one year after PTX [44,45,46].

## Data Availability

Data collected for this study will be made available by the corresponding author upon reasonable request.

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
