# Peer review of "Primary Hyperparathyroidism in Multiple Endocrine Neoplasia Type 2A in Denmark 1930–2021: A Nationwide Population-Based Retrospective Study"

_cancers, 2023, doi:10.3390/cancers15072125_

Round 1

Reviewer 1 Report

see file:

pHPT in MEN2

In a nationwide population-based retrospective study in MEN 2A the authors investigated the frequency of primary hyperparathyroidism (PHPT) in MEN2A patients in Denmark. In older studies PHPT is reported in up to 35% of affected individuals. In Denmark out of 204 MEN 2A cases, 16 had PHPT, resulting in a much lower frequency of 8%. There was also a lower age-related penetrance and a higher age at diagnosis of PHPT compared to the literature. This study is well done and the population-based analysis is new.

The conclusion that the prevalence of PHPT in MEN2A is lower than reported in the literature has to be discussed considering the specific distribution of the different RET mutations in Denmark.

In their cohort 94% of patients have RET mutations in exon 10, 13-15 (moderate risk), 10 of them have PHPT, while only 6% have a mutation in exon 11 (Codon 634 mutation, high risk), 6 of them have PHPT.

It is well known that different mutations in the RET gene are correlated with varying phenotypes of the disease, including the presence or absence of other endocrine neoplasms such PHPT (genotype - phenotype correlation). PHPT is mostly associated with codon 634 mutations (high risk) and less common in other RET mutations. In larger MEN2A series more than half of the patients with MEN2A have 634 mutations and therefore overall rate of PHPT is higher.

The authors revealed that they cannot rule out that the p.Cys611Tyr founder effect may lower their PHPT frequency. The low prevalence of PHPT in MEN2A in Denmark has to be qualitied in the context of the specific distribution of the RET mutations in Denmark. This has to be discussed more extensively.

Author Response

27-March-2023

Dear Assigned Editor Aiyarucht Techajindamanee and reviewers

We thank you for the careful and constructive review of our manuscript, and for the willingness to let us submit a revision of the manuscript for review in Cancers. The comments have been very helpful and have indeed improved the manuscript.

We have addressed the comments and the manuscript has been revised accordingly. Please find the reviewer comments and our responses below.

Best regards

Magnus Holm

REVIEWER 1

In a nationwide population-based retrospective study in MEN 2A the authors investigated the frequency of primary hyperparathyroidism (PHPT) in MEN2A patients in Denmark. In older studies PHPT is reported in up to 35% of affected individuals. In Denmark out of 204 MEN 2A cases, 16 had PHPT, resulting in a much lower frequency of 8%. There was also a lower age-related penetrance and a higher age at diagnosis of PHPT compared to the literature. This study is well done and the population-based analysis is new.

The conclusion that the prevalence of PHPT in MEN2A is lower than reported in the literature has to be discussed considering the specific distribution of the different RET mutations in Denmark.

We agree. New calculations have been added for frequency and age-related penetrance in ‘Results’ (lines 202-203 & 208-210) and the results are discussed in the ‘Discussion’ (lines 328-330 & 333-335).

In their cohort 94% of patients have RET mutations in exon 10, 13-15 (moderate risk), 10 of them have PHPT, while only 6% have a mutation in exon 11 (Codon 634 mutation, high risk), 6 of them have PHPT.

It is well known that different mutations in the RET gene are correlated with varying phenotypes of the disease, including the presence or absence of other endocrine neoplasms such PHPT (genotype - phenotype correlation). PHPT is mostly associated with codon 634 mutations (high risk) and less common in other RET mutations. In larger MEN2A series more than half of the patients with MEN2A have 634 mutations and therefore overall rate of PHPT is higher.

The authors revealed that they cannot rule out that the p.Cys611Tyr founder effect may lower their PHPT frequency. The low prevalence of PHPT in MEN2A in Denmark has to be qualitied in the context of the specific distribution of the RET mutations in Denmark. This has to be discussed more extensively.

Please see above, and please also see the attachment

Reviewer 2 Report

This retrospective evaluation had been done to describe the incidence of MEN2a in patients with hyperparathyroidism. 

This a nationwide study, going back to the year 1930 including a high number of patients. There a only a few comparable studies, and those  mainly were from single centres. 

queries:

1. It should be better explained, how the databank had been created?

2. Are all diagnoses of the Danish population available?

3. Because I assume that most of the readers are not familiar with the Danish health system, it should be shortly explained how it is possible to back nearly 100 years  

4.the gebetic tests had been done from frozen material?

Author Response

27-March-2023

Dear Assigned Editor Aiyarucht Techajindamanee and reviewers

We thank you for the careful and constructive review of our manuscript, and for the willingness to let us submit a revision of the manuscript for review in Cancers. The comments have been very helpful and have indeed improved the manuscript.

We have addressed the comments and the manuscript has been revised accordingly. Please find the reviewer comments and our responses below.

Best regards

Magnus Holm

REVIEWER 2

This retrospective evaluation had been done to describe the incidence of MEN2a in patients with hyperparathyroidism. 

This a nationwide study, going back to the year 1930 including a high number of patients. There a only a few comparable studies, and those mainly were from single centres. 

queries:

  1. It should be better explained, how the databank had been created?

A more extensive description has been added (lines 98-108).

  1. Are all diagnoses of the Danish population available?

Yes.

  1. Because I assume that most of the readers are not familiar with the Danish health system, it should be shortly explained how it is possible to back nearly 100 years

Along with the added more extensive description, this is explained in another paragraph (lines 144-150).

  1. the genetic tests had been done from frozen material?

Testing was not done from frozen material. In general, in the Danish MEN 2 cohort, testing was performed on genomic DNA extracted from whole blood. Only a few deceased cases were tested using formalin-fixed paraffin-embedded (FFPE) tissue. However, no patient in the current study was tested on FFPE tissue.

Reviewer 3 Report

The topic is interesting and the manuscript sheds light on an important side of a rare but disease group. The methodology is sound, the results are presented clearly and the manuscript is prepared with care.

Here are some minor comments and suggestions

Has the Danish RET cohort 2015-2021 been described in the literature? I did not find any published data. A more comprehensive description would be a good idea in case it has not been published previously. Has RET been sequenced in the whole Danish population (all inhabitants) or did the authors apply an approach similar to that of the 1974-2014 The Danish MEN2A Nationwide study?

The PHPT definition criteria that were applied by the authors are widely accepted but might miss a subgroup of the so-called subclinical or normocalcemic PHPT which would introduce an underestimation of the PHPT prevalence. Did the authors look into cases with elevated PTH but high-normal serum calcium levels?

Lines 405 to 407: It seems to me that the second sentence contradicts the first one. How is it possible that 40% of MTC/CCH were diagnosed after the PHPT if PHPT did not precede MTC in any of the patients?

Lines 414 to 420 Diagnosing two conditions in a certain sequence does not mean that they developed in the same order. Therefore, the reversed times of diagnosis would not overturn the biologically determined order. It might be assumed that clinical signs*symptoms (hypertension)  or routine lab tests (calcium) and not time of development will cause Pheo or PHPT to be diagnosed first in some cases.

Some repetitions of the results data in the discussion section are redundant and might be abridged.

Author Response

29-March-2023

Dear Editor-in-Chief and reviewers

We thank you for the careful and constructive review of our manuscript, and for the willingness to let us submit a revision of the manuscript for review in Cancers. The comments have been very helpful and have indeed improved the manuscript.

We have addressed the comments and the manuscript has been revised accordingly. Please find the reviewer comments and our responses below.

Best regards

Magnus Holm

The topic is interesting and the manuscript sheds light on an important side of a rare but disease group. The methodology is sound, the results are presented clearly and the manuscript is prepared with care.

Here are some minor comments and suggestions

Has the Danish RET cohort 2015-2021 been described in the literature? I did not find any published data. A more comprehensive description would be a good idea in case it has not been published previously. Has RET been sequenced in the whole Danish population (all inhabitants) or did the authors apply an approach similar to that of the 1974-2014 The Danish MEN2A Nationwide study?

We agree. The Danish RET cohort 2015-2021 has not been described earlier. A more comprehensive description has been added (lines 110-112).

The PHPT definition criteria that were applied by the authors are widely accepted but might miss a subgroup of the so-called subclinical or normocalcemic PHPT which would introduce an underestimation of the PHPT prevalence. Did the authors look into cases with elevated PTH but high-normal serum calcium levels?

We agree. Using the PHPT criteria based on a major international guideline, one might miss those with normocalcemic hyperparathyroidism. Unfortunately, we did not look into this as the diagnosis of normocalcemic PHPT in Denmark is controversial and at best is regarded as a very mild disease that does not require treatment.

Lines 405 to 407: It seems to me that the second sentence contradicts the first one. How is it possible that 40% of MTC/CCH were diagnosed after the PHPT if PHPT did not precede MTC in any of the patients?

We thank the reviewer for finding this error. It has been corrected (line 446).

Lines 414 to 420 Diagnosing two conditions in a certain sequence does not mean that they developed in the same order. Therefore, the reversed times of diagnosis would not overturn the biologically determined order. It might be assumed that clinical signs*symptoms (hypertension)  or routine lab tests (calcium) and not time of development will cause Pheo or PHPT to be diagnosed first in some cases.

We agree. The sentence has been changed for better understanding (lines 456-457).

Some repetitions of the results data in the discussion section are redundant and might be abridged.

We agree that there are some repetitions of our result in the discussion section. We have tried remove some of them. However, we also see it, as a service to the reader who does not have to scroll up to the results section every time, we compare a result to the literature.
